# Influence of Paste Strength on the Strength of Expanded Polystyrene (EPS) Concrete with Different Densities

**DOI:** 10.3390/polym14132529

**Published:** 2022-06-21

**Authors:** Diyang He, Wukui Zheng, Zili Chen, Yongle Qi, Dawang Zhang, Hui Li

**Affiliations:** School of Materials Science and Engineering, Xi’an University of Architecture and Technology, Xi’an 710055, China; he18149406571@163.com (D.H.); zheng.wukui@xauat.edu.cn (W.Z.); czl1234567892022@163.com (Z.C.); q18992952796@163.com (Y.Q.); zhangdawang1314@163.com (D.Z.)

**Keywords:** EPS concrete, surface fitting, compressive strength, binding material

## Abstract

Concrete in which EPS (expanded polystyrene) particles partially or completely replace concrete aggregates is called EPS concrete. Compared to traditional concrete, EPS concrete has a controllable low density and good thermal-insulation performance, which make it promising for prospective applications. At present, research on EPS concrete mostly focuses on increasing its strength and EPS surface modifications. Few researchers have studied the influence of cementitious material strength and EPS-concrete density on the strength of EPS concrete. In this research, cement was used as the main material, and fly ash, silica fumes, and blast furnace slag were selected as admixtures. By changing the mixing proportions of the admixtures, the basic properties, such as the paste strength, change. Based on the mix proportions of the above different raw materials, EPS concrete with different density levels was prepared to explore the influence of the density of EPS concrete and the strength of cementitious materials on the strength of EPS concrete. The influence of the slurry strength on EPS-concrete strength was weaker than that of the density of EPS concrete. When the strength range of the cementitious materials is 35.7~70.5 MPa, the compressive strength range of 1000 kg/m^3^, 1200 kg/m^3^, and 1400 kg/m^3^ EPS concrete is 8.8~17.6 MPa, 11.4~18.0 MPa, and 15.7~26.6 MPa, respectively. Based on the experiments, the fitting equation to determine the EPS-concrete strength–EPS-concrete density–cementitious material strength is z = 69.00087 + 0.0244x − 0.1746y − 0.00189x^2^ + 0.0000504706y^2^ + 0.00028401xy. Additionally, a strength-increasing design method for EPS concrete with different densities prepared by conventional Portland cement is clarified. This study can guide the preparation of EPS concrete.

## 1. Introduction

According to data from the International Energy Agency, about 30% of global final energy consumption belongs to the construction industry, and energy consumption has increased by 21% since 2000 [1,2]. A lot of energy is dissipated in the building operation stage. If the thermal-insulation performance of construction material can be improved, then the energy consumption of buildings can be greatly reduced [3,4]. 

Concrete is the most widely used material in building, but ordinary concrete has a large self-weight, low specific strength, and poor thermal-insulation performance [5,6]. In past decades, researchers have tried to use various aggregates to prepare lightweight concrete that can achieve good thermal-insulation performance [7,8]. In the 1950s, there were attempts to add EPS (expanded polystyrene) particles during the preparation of thermal insulation lightweight aggregate concrete [9,10]. This type of concrete was called EPS concrete [11,12].

Research on EPS concrete is mainly carried out from two aspects [13,14]. The first aims to improve the strength of the pastes to improve the strength of EPS concrete [15,16]. For example, Chen [17] and Peng [18] increased the compressive strength of EPS concrete by adding fly ash and blast furnace slag to EPS concrete. Ran [19] achieved the same results by adding micro silica fumes. In addition, Liu [20] and Guo [21] added polypropylene fibers, which increased the compressive strength of EPS concrete by 30%. The other method aims to improve the interface strength between EPS particles and cement slurry to improve the performance of EPS concrete [22,23]. For example, Xu [24] and Zhao [25] prepared higher strength EPS concrete after thermal and EVA modifications, respectively. The current research mainly focuses on how to improve the compressive strength of EPS concrete, and the compressive strength design of EPS concrete is also important in engineering practice. For the concrete mix design, it is common practice for concrete production to be based on a test mix ratio. To help the actual construction and mix design process and to minimize experimental tasks, the mathematical free model can be used in the form of a regression formula to predict the strength of a concrete mixture [26,27]. Meanwhile, the compressive strength of ordinary concrete mainly depends on the porosity of the hardened concrete, the strength of the cementitious materials, and the strength of the aggregate. However, unlike ordinary concrete, the aggregate of EPS concrete has a very small elastic modulus compared to other cementitious materials. Secondly, because the aggregate interface transition layer in EPS concrete affects the compressive strength of concrete to a large extent, it intensifies the complexity of predicting its performance [28,29]. In recent years, some researchers have carried out a series of studies. For example, A. Sadrmomtazi established a model to determine the effect of EPS particles and the cement content on the strength of EPS concrete [30]. Tuan established the EPS-concrete microstructure and strength change model [31]. J. Sobhani established a model of the relationship between the density and flexural strength of EPS concrete [32]. E. Akis established a model demonstrating the relationship between elastic modulus and compressive strength [33]. Through various experiments, Nadhim proved that the strength of cementitious materials greatly affects the strength of EPS concrete [34]. In summary, the current modeling research generally infers the compressive strength of EPS concrete through the physical properties of EPS concrete, which is cumbersome and inconvenient in practical construction. For EPS concrete, there are two main factors affecting the compressive strength, namely, the density of the EPS concrete and the strength of the cementitious materials [35]. Therefore, the establishment of a cementitious material strength–EPS-concrete density–EPS-concrete strength model can greatly facilitate the application of EPS concrete in actual construction [36,37].

In this paper, the compressive strength of pastes is changed by changing the type and amount of the mineral admixtures [38,39]. By preparing EPS concrete with different densities, the relationship between pastes with different compressive strengths and EPS concrete with different densities is comprehensively explored, and the fitting formula between the strength of pastes and the density of EPS concrete on the strength of EPS concrete is established. 

## 2. Materials and Methods

### 2.1. Raw Materials

The cement used in the experiment was PO (ordinary Portland cement) 42.5 produced by Conch Cement (Xiaan, China). EPS pellets were provided by Yiwu mianer Crafts Co., Ltd. (Zhejiang, China) with particle size of 0.5~1 mm, a bulk density of 18~21 kg/m^3^, and an apparent density of 33 kg/m^3^. The fly ash came from Sanmenxia thermal power station in Henan Province (Sanmenxia, China), S105 blast furnace slag and silica fumes were purchased from Hebei Yousheng refractory Co., Ltd., (Shijiazhuang, China). Their chemical composition is shown in Table 1. Tap water was used for the mixing water.

### 2.2. Experimental Method and Mix Proportion Design

The experimental method is shown in Figure 1. The paste strength was changed via the use of three mineral admixtures with different amounts of fly ash, silica fumes, and blast furnace slag, and EPS-concrete samples with densities of 1000 kg/m^3^, 1200 kg/m^3^, and 1400 kg/m^3^ were configured. After changing the mineral admixtures, the changes in the paste test block and the corresponding EPS-concrete strength were tested.

In total, 15 groups of pastes with different strengths were used in the experiments. The mix proportions of the pastes are shown in Table 2. According to the calculation to determine the absolute volume, 23.3 kg, 15.6 kg, and 10.3 kg of EPS particles were added to the pastes to prepare EPS-concrete samples with densities of 1000 kg/m^3^, 1200 kg/m^3^, and 1400 kg/m^3^, respectively.

### 2.3. Experimental Process and Test Method

During the experiment, the pastes were weighed according to the mix proportions and poured into the horizontal mixer for mixing; then, the pastes were pre-mixed for 2 min to ensure that the dry powder was evenly mixed; this was followed by adding 2/3 of the water and all of the EPS particles. After stirring for 3 min, the other 1/3 of the water was added, and the mixture was stirred for another 3 min. Finally, the samples were poured into 1000 × 1000 × 1000 mm molds. The curing and the tests were performed according to Standard JC/T 2458-2018 (polystyrene granule foamed concrete) [40].

The compressive strengths and dry densities of the test blocks were tested according to the national standard JC/T 2458-2018 (polystyrene crosslinked polystyrene concrete) [40]. A test block was placed in the compressive strength tester (Ken Machinery Electronics Co., Ltd. YES-2000B, Shaoxing, China) and then continuously loaded at a rate of 1.5 MPa until the test block was destroyed. The compressive strength value was recorded, and the next test block was tested. In total, six test blocks were measured for each group, and the results were averaged. The dry density can only be measured after concrete reaches a certain age. For the measurements, the lengths and widths of the test block were measured with a ruler, and then the test blocks were placed in an oven at 55 °C. The masses of the test blocks were measured every 4 h. When the weight difference between the two measurements was less than 1 g, the average value of the two measurements was taken as the mass of the test block. Then, the density of the test blocks was calculated according to the mass and volume of each volume. In total, six test blocks were measured for each group, and the results were averaged.

FTIR was used to determine whether the chemical bonds in the test blocks had changed. This was because, in addition to the EPS content, each group had the same proportions, so 28 d was selected as the aging period for the EPS-concrete samples with a density of 1000 k/m^3^. The steps for creating the concrete were as follows: first, the EPS-concrete test blocks were destroyed, and a certain number of samples from the middle were obtained. All of these samples were broken by being pushed through an 80 μm standard sieve. Finally, the samples were dried at 60 °C, and the Nicolet IS5 Fourier transform infrared spectrometer produced by the US company Thermo Fisher Scientific was used. The parameters were as follows: resolution 4 cm^−1^, scanning frequency 16 times per minute, and scanning range 400~4000 cm^−1^

## 3. Result and Discussion

The XRD patterns of fly ash, mineral powder, and silica fume are shown in Figure 2. Fly ash contains quartz and mullite. Mineral phase peaks that can be used for analysis could not be found between mineral powder and silica fume.

The influence of the paste strength on the strength of EPS concrete under different EPS-concrete densities is shown in Figure 3. The strength range of 7 d age cementitious material is 35–60 MPa, and 28 d age cementitious material strength range is 45–70 MPa. With the increase in age, the strength range of cementitious materials also increases. As seen from the figure, the compressive strengths of pastes range from 35.7 MPa to 70.5 MPa. When the density of the EPS concrete is 1000 kg/m^3^, 1200 kg/m^3^, and 1400 kg/m^3^, the compressive strength increases from 8.8 MPa to 17.6 MPa, from 11.4 MPa to 18.0 MPa, and from 15.7 MPa to 26.6 MPa, respectively. The growth trend for strength is also different at different densities. Compared to the 1000 kg/m^3^ EPS concrete, the increasing trend of the 1200 kg/m^3^ and 1400 kg/m^3^ concrete sample with the increase in the paste strength is more obvious. Therefore, the lower the density of EPS concrete, the more difficult it is to improve its compressive strength.

The paste strength of the 1000 kg/m^3^, 1200 kg/m^3^, and 1400 kg/m^3^ of EPS concrete was fitted with the power function for the strength of EPS concrete. The fitting curves with different densities are shown in Figure 3. It can be seen from the figure that, with the increase in density, the high-density fitting curve is above the low-density fitting curve. The EPS-concrete fitting curve of the 1000 kg/m^3^ and 1200 kg/m^3^ EPS concrete is smoother than that of the 1400 kg/m^3^ concrete. Therefore, when the density of EPS concrete is lower than 1000 kg/m^3^, the extreme value of the curve will continue to decrease and will become smooth earlier. When the density of EPS concrete is higher than 1400 kg/m^3^, the extreme value of the curve will be higher, and it will become gentler when the strength of paste is greater than 70 MPa. In addition, it can be seen that, with an increase in the density of EPS concrete, the fitting correlation also increases because, as the paste amount increases, the strength of the EPS concrete is more similar to that of the paste strength. When only the compressive strength data after 7 d is used, the range of the fitting formula is 35–60 MPa, and the range of the fitting formula is extended to 35–70 MPa after adding 28 d age data. This is because the more data points there are at each density, the higher the accuracy of the fitting formula is. At the same time, the data of compressive strength of 7 d age and 28 d age can make the fitting formula closer to the actual situation. The fitting formula is listed in Table 3. 

Some scholars have shown that the stress distribution of cement-based materials depends on the particle size, paste, and the relative elastic modulus of the particles. When the elastic modulus of the aggregate is higher than that of the paste, then the stress is concentrated near the aggregate [41,42]. However, for EPS concrete, the elastic modulus of the aggregate is very small compared to that of the paste and can be ignored. Therefore, EPS particles can be regarded as holes or defects in the concrete [43], and when the strength of EPS concrete is increased to a certain extent, the defects in the EPS particles greatly affect the increase in the EPS-concrete strength, resulting in the gradual slowdown of the strength improvement rate, especially at densities of 1000 kg/m^3^ and 1200 kg/m^3^. As for increases in density, the 1400 kg/m^3^ EPS concrete still shows an increasing trend because the smallest amount of EPS particles was added, and the defects are relatively small. 

Figure 4 and Figure 5 show the FTIR spectra of 28-day-old EPS concrete with a density of 1000 kg/m^3^. When the wave number is 3643 cm^−1^, it represents the stretching vibration peak of free hydroxyl, indicating the difference in water content. The peak value of the wave number is in the range of 3200~3500 cm^−1^ and represents hydroxyl, indicating the Ca(OH)_2_ content in the test block. The wave number of 1633 cm^−1^ represents the active carbon–carbon double bond, and this peak represents the EPS particles in the sample [44]. Due to the same mixing ratio being used, the content of the EPS particles in each group of samples is also the same. The wave number of 1434 cm^−1^ represents the C-O bond. According to the literature, this peak represents the degree of erosion in cement hydration products by CO_2_ [45]. The peaks with a residual wave number at 983 cm^−1^ and 463 cm^−1^, respectively, represent the stretching vibration deformation of the Si-O bond and Si-O. The number of Si-O bonds represented by the wave number at 983 cm^−1^ indicates the C-S-H content in hydrates, and there is little difference between each peak. This shows that there is little difference in the hydration products generated by each group of mixture ratios, and the changes in the strength of EPS concrete are mainly due to the effects of volcanic ash and micro-aggregates in various solid-waste products.

Researchers Daneti [46] integrated the compressive strength and density data of lightweight high-strength EPS concrete prepared by other researchers and inverted their data into a formula: ƒ_c_ = 10.3 × γ^1.918^ × 10^−6^ (ƒ_c_ represents the strength of EPS concrete in MPa, and γ represents the density of EPS concrete in kg/m^3^). In the formula, the strength of the EPS concrete increases as the density increases, but the changes in the strength of the EPS concrete under different densities are not considered. In order to comprehensively consider the influence of the compressive strength of paste and the EPS-concrete density on the compressive strength of EPS concrete [47], Figure 6 shows the surface fitting equation and an image of the cementitious material strength–EPS-concrete density–EPS-concrete strength.

As seen from Figure 6, the strength of EPS concrete also shows an increasing trend as the EPS-concrete density or paste strength increases. The maximum strength of EPS concrete is 24 MPa when the density of the EPS concrete is 1400 kg/m^3^ and the strength of the paste is 70 MPa. The lowest point of EPS-concrete strength is in the opposite direction to the highest point and can be observed when the density of the EPS concrete is 1000 kg/m^3^ and the strength of the paste is 30 MPa. This is also consistent with experimental results. The surface fitting formula in Figure 6 can be used to calculate the target strength of EPS concrete using the raw materials and density. According to the conclusions of existing research, the strength of concrete increases as the strength of the cementitious materials and EPS-concrete density EPS increase. However, one coefficient of x and y in the fitting formula is negative, respectively. According to the calculation, it can be concluded that the partial derivatives of x and y to z are greater than 0 in the scope of the formula, which is sufficient to show that x and y increase monotonically relative to z. This means that, as the density of EPS concrete and the strength of the cementitious materials increase, the strength of EPS concrete also increases. The positive and negative coefficients only indicate the change rule for x or y and do not indicate that x or y shows a decreasing trend relative to z.

The surface fitting formula is depicted as a contour line in Figure 6 on the X–Y plane, as shown in Figure 7. Each line represents the value required by the paste strength and the density of EPS concrete when the strength of EPS concrete reaches a certain value. In practice, ordinary Portland cement with strength numbers of PO 32.5, PO 42.5, and PO 52.5 are commonly used to achieve lightweight EPS concretes C5, C10, C15, C20, and C25. Therefore, the strengths that EPS concrete can reach when prepared at different densities with existing cement are highlighted in Figure 7. By querying this figure, the density and strength of EPS concrete can be obtained directly. For example, EPS concrete prepared with PO-32.5-strength cement and a density of 1400 kg/m^3^ is used has a compressive strength 15 MPa. Moreover, if EPS concrete were prepared using PO 32.5 cement and a density of 1300 kg/m^3^, its strength would be 10 MPa.

In addition, it can also be seen from the figure that, when preparing EPS concrete at a given density, it is difficult to achieve the goal of only using cement as paste. By adding a highly active mineral admixture, the density can be reduced, and the strength can be improved. At the same time, according to the surface fitting formula, the density of EPS concrete and the paste strength need to be considered at the same time when determining the strength of EPS concrete. It is difficult to improve the strength of EPS concrete by simply increasing the paste strength. It is simpler to improve the strength of EPS concrete by increasing the density. The formula that was fitted in this study can calculate the strength of EPS concrete according to the paste strength and the density of EPS concrete. It provides a basic method for the economical and reasonable preparation of EPS concrete. 

## 4. Conclusions

In this study, EPS concrete with three different densities were prepared using pastes with different properties. The relationship between the strength of EPS concrete and paste strength was explored, and the following conclusions were drawn:

(1) According to the fitting, the relationship between the paste and the strength of EPS concrete is a power function with an index less than 0 and in the range of 30 MPa to 70 MPa, and the increase in the strength of EPS concrete gradually slows down as the paste strength increases.

(2) The binary fitting formula for the EPS concrete density and paste relative to EPS concrete strength is:$$z=69.00087+0.0244x-0.11746y-0.00189x^{2}+0.0000504706y^{2}+0.00028401xy$$

The formula can calculate the target EPS-concrete strength according to the raw materials and density used for preparation.

(3) When preparing EPS concrete, the use of a mineral admixture can reduce the density of EPS concrete and improve its compressive strength. When using higher grade cement with a lower density, the strength of EPS concrete cannot be improved significantly.

In terms of future research, it still is necessary to prepare EPS-concrete samples with other densities to modify this formula and to increase the scope of its application. In addition, the smaller the density interval of EPS concrete is, the more data is used, and the more accurate the fitting formula is. This will be a big challenge. At the same time, it is also necessary to increase the fitting formula of EPS concrete under other paste strengths. If EPS concrete with a density from 200 kg/m^3^ to 1800 kg/m^3^ is prepared, in future practice or scientific research, the strength of EPS concrete with a given density can be inferred by knowing the paste strength, facilitating construction practices. The workload in mix design will be greatly reduced, and the applicable scene of EPS concrete will also be improved.

## Figures and Tables

**Figure 1 polymers-14-02529-f001:**
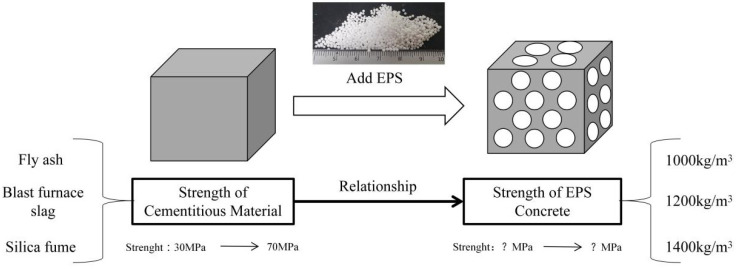
Experimental method.

**Figure 2 polymers-14-02529-f002:**
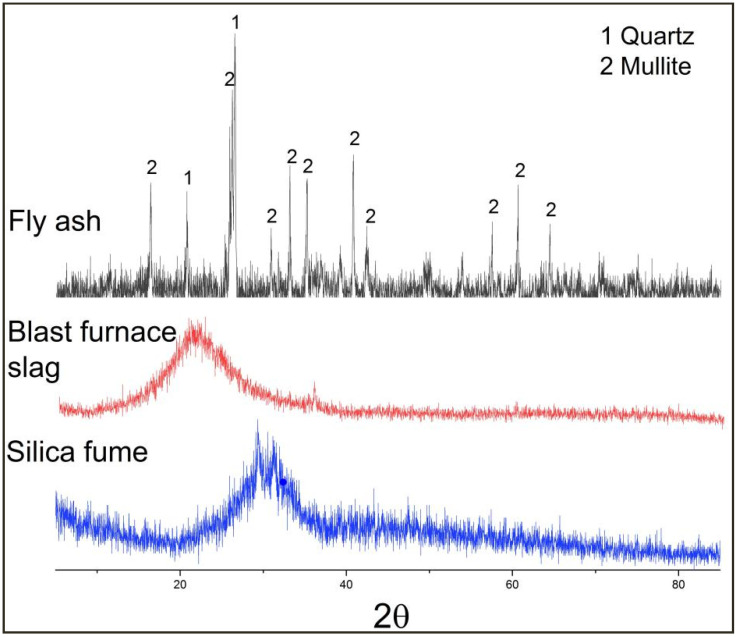
XRD analysis of fly ash, blast furnace slag, and silica fumes.

**Figure 3 polymers-14-02529-f003:**
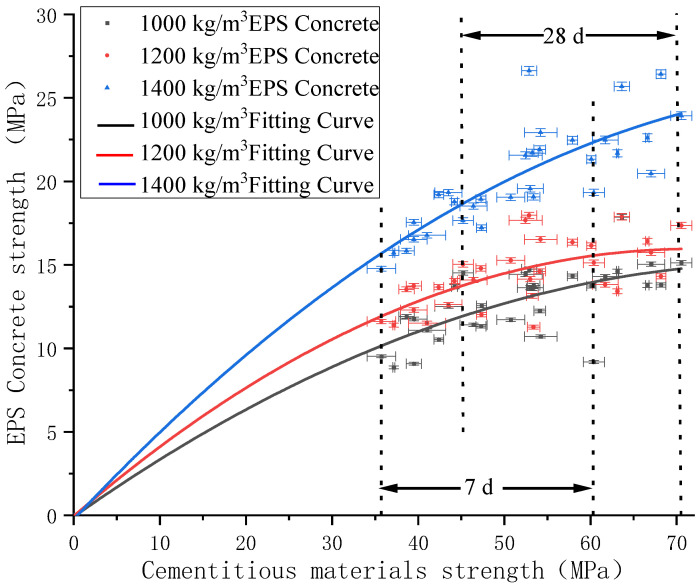
Fitting diagram of the effect of paste strength on EPS-concrete strength.

**Figure 4 polymers-14-02529-f004:**
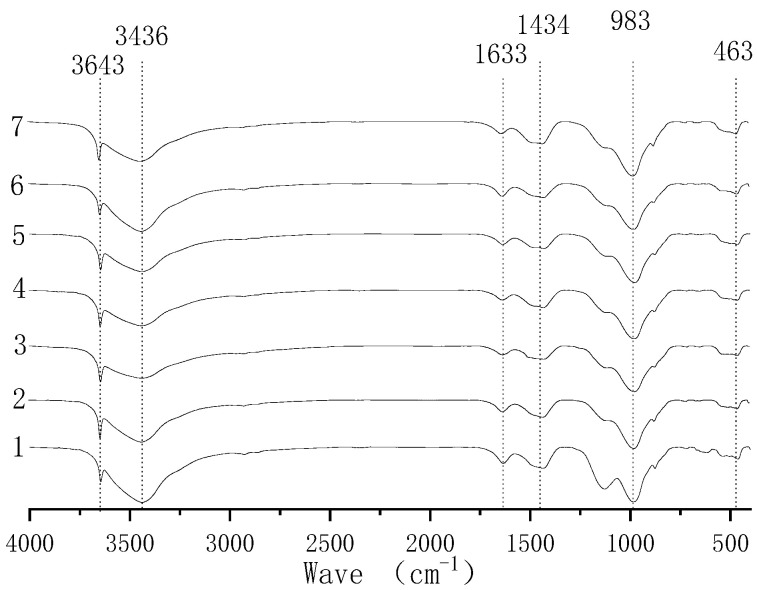
FTIR analysis of EPS concrete in groups 1–7.

**Figure 5 polymers-14-02529-f005:**
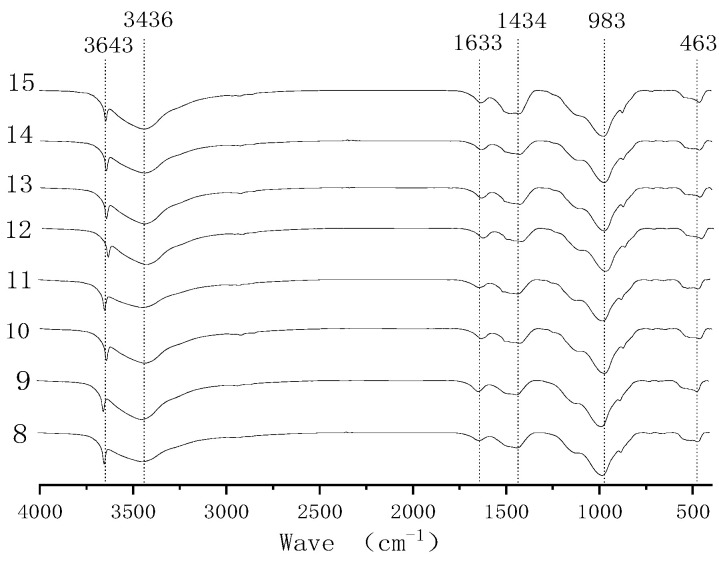
FTIR analysis of EPS concrete in groups 8–15.

**Figure 6 polymers-14-02529-f006:**
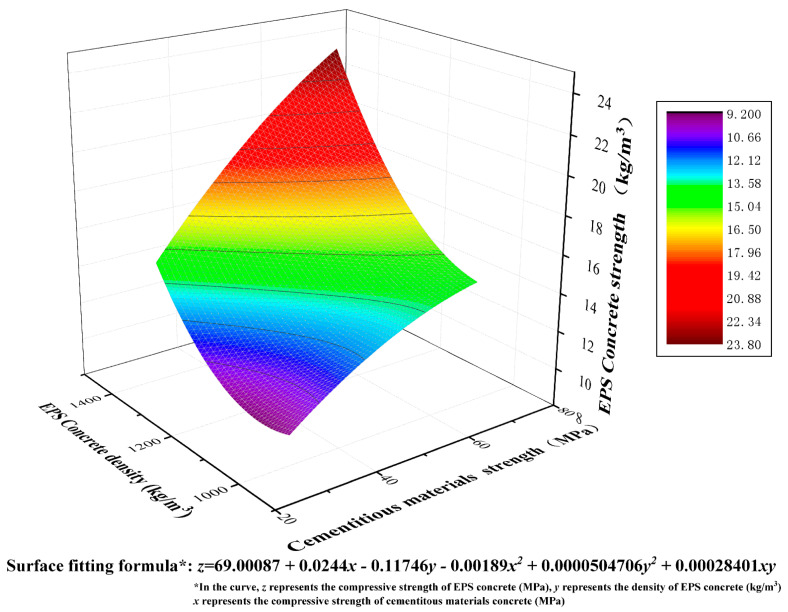
Fitting curve for paste strength and EPS-concrete strength.

**Figure 7 polymers-14-02529-f007:**
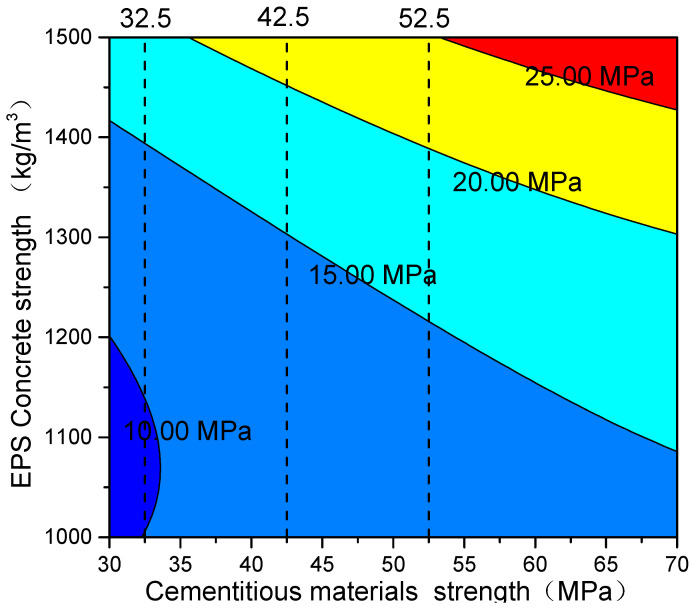
Contour map of the surface fitting map projected on the X-Y plane.

**Table 1 polymers-14-02529-t001:** Chemical composition of fly ash, blast furnace slag, and silica fumes.

Chemical Composition (wt.%)	SiO_2_	Al_2_O_3_	Fe_2_O_3_	MgO	CaO	Na_2_O	K_2_O	Others
Fly ash	47.1	33.1	4.2	0.9	3.9	0.8	1.9	8.1
Blast furnace slag	58.7	15.3	3.9	0.80	3.1	3.1	5.9	9.2
Silica fume	91.2	0.6	0.1	0.7	0.5	0.5	1.4	5.0

**Table 2 polymers-14-02529-t002:** Mix proportion of the pastes.

Experimental Groups	Cement (kg)	Fly Ash (kg)	Silica Fumes (kg)	Blast Furnace Slag (kg)	Water (kg)
1	1000				400
2	900	100			400
3	800	200			400
4	700	300			400
5	600	400			400
6	980		20		400
7	960		40		400
8	940		60		400
9	920		80		400
10	900		100		400
11	850		150		400
12	900			100	400
13	800			200	400
14	700			300	400
15	600			400	400

**Table 3 polymers-14-02529-t003:** Fitting curve of paste strength on EPS concrete.

EPS-Concrete Density	Fitting Curve *	R-Squared
1000 kg/m^3^	$y=19.0479-19.1783\times e^{\frac{-x}{45.4151}}$	0.86
1200 kg/m^3^	$y=18.5339-18.5677\times e^{\frac{-x}{33.9987}}$	0.88
1400 kg/m^3^	$y=34.6958-34.8802\times e^{\frac{-x}{58.2959}}$	0.91

* In the curve, *y* represents the compressive strength of EPS concrete and *x* represents the compressive strength of paste.

## Data Availability

Data are available upon request from the corresponding author.

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
