# Peer review of "Influence of Paste Strength on the Strength of Expanded Polystyrene (EPS) Concrete with Different Densities"

_polymers, 2022, doi:10.3390/polym14132529_

Round 1

Reviewer 1 Report

Article Title

Influence of Paste Strength on the Strength of EPS Concrete with
Different Density

  1. The study objectives are not clear. Please provide a more comprehensive detail highlighting the objectives concerning this study. 
  2. The abstract must be revised with the quantitative results.
  3. The introduction is too short and unable to provide sufficient literature review to highlight the importance of this study.
  4. The authors must provide details of abbreviations before their use. For example PO.
  5. The experimental program is not well written. Why there are headings 1.2 after 2.1
  6. The research of some scholars shows that the stress distribution of cement-based materials depends on the particle size, paste and the relative elastic modulus of particles.
  7. The authors must provide proper references for each statement related to the previous studies.
  8. In order to comprehensively consider the influence of paste compressive strength and EPS concrete density on EPS concrete compressive strength[31], take EPS concrete density as the X-axis,
    paste strength as the Y-axis, and EPS concrete strength as the Z-axis, carry out surface fitting on the drawing results. The results are shown in Figure 5.
  9. The sentence is not clear. Please revise properly.
  10. There is a need to compare the results with the existing studies.
  11. Please concise conclusions. 

Reviewer 2 Report

The paper "Influence of Paste Strength on the Strength of EPS Concrete with Different Density" deals with the use of Expanded Polystyrene in concrete, evaluating its properties. The topic is interesting to potential readers, however as the paper is structured and presented it is not currently accepted, authors should focus heavily on reviewing this paper before resubmission:

(1) The title must not use acronyms;
(2) The abstract does not bring the innovation of this research, not even the main quantitative results found in its findings;
(3) The paper in general is outside the MDPI publication standard, this should not even be allowed;
(4) Literature review is weak and inconsistent. There are numerous researches related to the use of EPS in cementitious materials that are not even addressed. This is observed in only 31 references, many of them old, this cannot be allowed!
(5) Are the authors unable to contextualize which real innovation of their research and what does it bring new to potential readers?
(6) Both the experimental program and the discussion of the results denote a preliminary and initial research, not compatible with Polymers. Further discussion is lacking and here again the issue of limited theoretical framework is a problem.

Due to the problems exposed above, I indicate the rejection of this paper in its current state.

Reviewer 3 Report

The work analyzes the combined effect of different type/amount of admixtures and of different density of expanded polystyrene concrete on the strength of the final material. The reported results could be of interest for the skilled readers: thus, even if of scholastic quality, by way of encouragement for Authors I suggest to consider it for publication, but only after mandatory strong improvements.

- The main aim of the work is clear, but at least a few information about the materials properties would increase the scientific level of the paper – some hints: density and strength measurement details, morphology analysis (SEM of fracture surface) to enforce the (probably correct) observations about bulk holes and defects, etc.  

- Without further insights, the experimental plan sounds as testing 15 different pastes for 3 types of EPS concrete in term of density, thus I expected 45 experimental points in Fig.3, 15 for each color (by the way, there are two red squares, adjust color), whereas there are 90. More specifically, 15 paste strength values (on x-axis) should be related to three points (each for color) aligned on their vertical: as I can see there are 22 x-points with their own triplet, and even four x-points with a couple of related y-values for each color (22*3+4*12=90)…please clarify.

- Given the Fig.4, Fig.3 should be deleted, since no further info is given. The same for the fancy 3D plot of Fig.5, since the quadric surface does not give more details than what clearly shown in Fig.4 (which is much more readable). The relative formula, instead, is good for predictive aim – as well as the contour plot of Fig.6 – but, about this, I always suggest to give somewhat physical interpretation of the best fit regression results, in order to gain knowledge of the physical issue under investigation (not leaving everything to mere computation…): here, for example, how could be explained the negative coefficient of the quadratic x term (while, from everything was reported and claimed, the EPS concrete strength should increase monotonically with paste strength)???    

- Above all, language and syntax are very poor – a mother tongue revision is warmly suggested. Check also the sections numbering (2.2, 2.3).

Round 2

Reviewer 2 Report

There was a significant improvement in the quality of the paper by the authors, this is commendable, however some points still deserve attention and corrections:

(1) The literature needs to be complemented with some recent studies in the area, such as: 10.1016/j.cscm.2022.e00919; 10.1016/j.cscm.2021.e00804; 10.3390/ma14133549.

(2) There are characterization results that need to be inserted in the results section, not methodology as it is;

(3) The conclusion is still not good, the authors should dedicate themselves to showing future perspectives and challenges of this experimental program.

Reviewer 3 Report

Commendable efforts made to improve the scientific quality of the paper, nevertheless I still cannot hold it suitable to be considered for publication.

About the previous report: 

ANS:

“…based on the 45 groups of tests, the 7d and 28d compressive strength of each test block in the 45 groups of tests was measured. … there were some similar compressive strengths in the mixture ratios, resulting in some x values in Figure 3 corresponding to more y values. If the compressive strength of each group of cementitious materials varies greatly, an x value will only correspond to the y value of the EPS concrete samples with three different densities.”

Very, very cumbersome…to say the least! My humble hints:

- analyze 7d and 28d cases separately;

- arrange x values without overlay;

- report inter-blocks variations by error bars.

ANS:

“… we want to discuss the data themselves separately from the linear fitting, so Figures 3 and Figure 4 were made specifically for this purpose. The quadratic surface fitting diagram in Figure 5 can simultaneously reflect the increasing trend in the strength of EPS concrete with the density of the EPS concrete and the strength of the cementitious materials and the possible change trend observed afterward…. The partial derivative of x or y relative to z is greater than 0. …”

Again:

- Fig.3 is redundant;

- missed point (i.e., meaning of ‘physical interpretation): how could be explained the negative coefficient of the quadratic x term (which will dominate at higher and higher cementitious compressive strength) ?

Round 3

Reviewer 2 Report

The version that the reviewer had access to did not present the necessary corrections indicated above, and it was not possible to approve this version of the paper.

Reviewer 3 Report

Impossible to evaluate the revision – please provide the final version of the paper.

Round 4

Reviewer 2 Report

Now the authors have submitted all corrections, the paper can be accepted.

Reviewer 3 Report

The paper was improved, it might be now considerable for publication. Provide a fine further language revision, avoid to report the specific expression of the binary fitting within abstract and conclusion.